# Prevalence and associated factors of post-traumatic stress disorder among women with an experience of intimate partner violence (IPV): Insights from urban slums of Bangladesh

## Research Article

post-traumatic stress disorder; intimate partner violence; determinants; women; Bangladesh

**Corresponding author:**
Kamrun Nahar Koly;
Email: koly@icddrb.org

Kamrun Nahar Koly[1] [iD], Sanjida Sultana[2], Jobaida Saba[1], Maliha Khan Majlish[3], Md. Arif Billah[1], Juliet Watson[4] and Barbara Barbosa Neves[5]

[1]Health System and Population Studies Division, ICDDR, Dhaka, Bangladesh; [2]School of General Education, BRAC University, Dhaka, Bangladesh; [3]Department of Public Health and Hospital Administration, National Institute of Preventive and Social Medicine, Dhaka, Bangladesh; [4]Social Equity Research Centre, School of Global, Urban and Social Sciences, RMIT University, Melbourne, VIC, Australia and [5]Sydney Centre for Healthy Societies, School of Social and Political Sciences, Faculty of Arts and Social Sciences, The University of Sydney, Camperdown, NSW, Australia

## Abstract

Despite high rates of intimate partner violence (IPV) among women, research on its mental health consequences, particularly PTSD in slum settings, remains scarce. This study assessed PTSD prevalence and determinants among slum-dwelling women in Bangladesh who experienced IPV during the COVID-19 pandemic. A cross-sectional study was conducted between July and October 2022 among 291 women from 5 urban slums in Dhaka, who reported IPV using the World Health Organisation questionnaire. Face-to-face interviews collected socio-demographic data, pandemic-related challenges, gender roles, health information and PTSD symptoms using the validated Post-Traumatic Stress Disorder Checklist-5. Logistic regression identified PTSD predictors. Most women were married before the age of 18 years (87.9%), unemployed (69.3%), had no formal schooling (38.6%) and lived in overcrowded households (38.6%). Over half of their husbands were daily wage earners (57.9%) and had a history of substance misuse (65.9%). PTSD prevalence was 21.16% and was higher among women with non-communicable diseases (adjusted odds ratio [AOR]: 3.29; 95% confidence interval [CI]: 1.6–6.7), concern about COVID-19 infection (AOR: 3.87; 95% CI: 1.12–13.22) and increased marital arguments (AOR: 3.00; 95% CI: 1.57–5.74). IPV in slum settings imposes a significant PTSD burden, highlighting the need for community-based mental health services to support marginalised women.

## Impact statement

This study provides critical insights into the mental health consequences of intimate partner violence (IPV) on women living in urban slums, a marginalised setting where mental health conditions are neglected. In Bangladesh, over 3.4 million people live in slums, and this study addresses the emotional challenges of women whose daily lives are shaped by poverty, overcrowding, early marriage and limited healthcare access. In such a precarious state, the scars of IPV deepen, leaving women especially vulnerable to post-traumatic stress disorder (PTSD). Our findings reveal a stark reality: nearly one in four women, exposed to IPV in slum settings, experienced PTSD. The burden was even greater among women living with chronic health conditions, fearing coronavirus disease 2019 infection or facing heightened conflict with partners during the pandemic, illustrating the syndemic nature of IPV where structural inequities, illness and crisis intersect to magnify psychological harm. This research has the potential to inform public health policies and programs for promoting and safeguarding women's mental health through integrating accessible trauma-informed care into community health services and raising community awareness about gender-based violence. Additionally, stronger legal protections, expanded social safety nets and gender-inclusive community engagement are essential to break cycles of violence and silence about both IPV and mental health. Furthermore, it recommends incorporating interdisciplinary action to strengthen women's protection of their human rights. By scientifically endorsing the hidden psychological toll of IPV in one of the most disadvantaged urban populations, this study contributes evidence that is both locally grounded and globally relevant, underscoring the recognition and response to the mental health consequences of violence for advancing gender equity and resilient health systems.

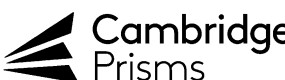

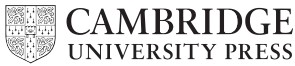

## Introduction

Intimate partner violence (IPV), according to the World Health Organisation (WHO), is the act of physical aggression, sexual coercion, psychological abuse and controlling behaviours that cause physical, sexual or psychological harm (Garbarino et al., 2021; WHO, 2024). IPV is a breach of human rights and occurs within international and local contexts of gender inequality, structural disadvantage and oppression. Globally, physical and/or sexual violence perpetrated by men against women's intimate partners is highly prevalent, predominantly among women from developing countries (WHO, 2005; Ellsberg and Emmelin, 2014; Sardinha et al., 2022).

In Bangladesh, IPV cases are common (42–76%) among women living in urban areas, especially those who are rural migrants living in slums (Naved and Persson, 2005; Sambisa et al., 2010; Sambisa et al., 2011; Parvin et al., 2016; Stake et al., 2020). According to the World Bank, ~52% of Bangladesh's urban population, about 1.8 million people live in slums, having migrated from rural to urban areas. This demographic shift eventually makes underprivileged individuals more prone to increased stress and social challenges, such as unemployment and housing scarcity, leading to altered family dynamics, social structures, behaviour and relationship patterns. Additionally, these migratory challenges influence the poor mental health and socio-cultural negligence of women from low-income communities, such as slums (Amjad, 2019; Agarwal et al., 2020). Furthermore, low socio-economic status and decision-making power, early marriage, limited access to healthcare and educational resources, financial insecurities and patriarchal culture contribute to the mistreatment of women by their own partners, leaving them 15% more vulnerable to IPV than those who are not living in similar marginalised settings (Sharmin and Luna, 2015; Al Helal et al., 2017; Sharma et al., 2020; Ralli et al., 2021). The recent coronavirus disease 2019 (COVID-19) pandemic deepened these chronic vulnerabilities, causing a dramatic rise in IPV cases in Bangladesh (Buttell and Ferreira, 2020; John et al., 2020; Abir et al., 2021; Agüero, 2021; Koly Islam et al., 2021; Koly et al., 2021; Wake and Kandula, 2022; Yunitri et al., 2022). Also, global studies documented that extended isolation, household economic challenges and limited access to support services resulted in increasing mental health disorders such as PTSD for victims of IPV (Bradbury-Jones and Isham, 2020; John et al., 2020; Mazza et al., 2020; Thibaut and van Wijngaarden C, 2020).

PTSD is a complex and long-term mental health condition and has been associated with IPV in different studies (Kemp et al., 1991; Dutton, 1995; SILVA et al., 1997). PTSD is brought on by going through traumatic life events and includes symptoms such as reliving and recollecting the event, depressive thoughts, self-destructive behaviour and prolonged psychological distress (DSM, 1994; Dutton, 1995). Women exposed to IPV are twice as likely to develop PTSD, with prevalence estimates among IPV survivors ranging from 30 to 50% with factors such as psychological abuse, chronic fear and limited coping mechanisms increasing vulnerability (Nathanson et al., 2012; Fernández-Fillol et al., 2021; Dai et al., 2023; White et al., 2024). Although mental health conditions among women with IPV experiences were investigated across pregnant, rural and urban populations, there is no empirical evidence of PTSD in female slum residents during the COVID-19 pandemic in Bangladesh (SILVA et al., 1997; Nasreen et al., 2011; Ziaei et al., 2016; Islam et al., 2017; De and Murshid, 2018; Jain et al., 2018; Esie et al., 2019; Rashid Soron et al., 2021; Tasnim et al., 2023). Therefore, there was a scope for exploring the interaction of COVID-19-related stressors with IPV and PTSD in urban slum settings, which is already defined by unstable living conditions and chronic insecurity. Moreover, no prior Bangladeshi study has applied a validated PTSD assessment tool, such as the PCL-5, in this population, limiting comparability with international research. Thus, this study is the first ever to aim to determine the prevalence of PTSD and the associated factors among women who have experienced IPV and live in urban slums in Dhaka, Bangladesh. Findings show a significant prevalence of PTSD among women living in precarious circumstances, such as urban slums (Chowdhury et al., 2018). It is, therefore, essential to design community-based interventions for PTSD targeting women who have unstable living circumstances and incorporate interdisciplinary action aimed at strengthening women's protection of their human rights.

## Methods

### Study setting

The study focused on the slum population in Dhaka City Corporation, where ~4 million people inhabit more than 5,000 slums. People living in slums face risks of crime, violence, limited health benefits and psychological distress due to densely populated settings, poverty and lack of formalised support. This study utilised the sample frame from the Urban Health and Demographic Surveillance Systems (Urban HDSS) of the International Centre for Diarrhoeal Disease Research, Bangladesh (icddr,b), which was conducted in 13 large and stable slums across 5 locations in Dhaka city. The urban HDSS covers Korail, Mirpur (Dhaka North City Corporation), Shyampur, Dholpur (Dhaka South City Corporation) and Tongi (Gazipur City Corporation), where ~1,56,000 individuals reside in nearly 40,000 households. Most dwellings (81.6% of households) have only one room, and 90% of them share a single toilet and a piped water source. About half (50%) of residents dispose of their daily garbage in an open space just outside their homes (Razzaque et al., 2023).

### Study design and population

This cross-sectional study was conducted between July and October 2022. The study participants were selected by employing the following inclusion criteria: (i) enrolled in Urban HDSS, (ii) aged between 18 and 60 years old, (iii) married, (iv) lived with husband for the last 2 years before data collection (during the entire period of the pandemic) and (v) lived in the respective slum for the years preceding the study. Critically ill and pregnant women were excluded from the study due to added ethical issues, potential health risks and the need for specialised care that could affect the study outcomes. Also, pregnancy is a special period of time with several hormonal and physiological changes, and this population is vulnerable and prone to antenatal depression. This study adhered to the criteria of Strengthening the Reporting of Observational studies in Epidemiology (STROBE) (Supplementary 1).

The sample size was calculated by using the following standard formula (1), where $n$ is the required sample size, $z$ is the $z$-value for the desired confidence level, $p$ is the estimated population proportion (0.5 for maximum variability), $q = 1 - p$ and $d$ is the margin of error.

$$n = \frac{z^2 pq}{d^2} \qquad (1)$$

Considering the 40% prevalence of depressive symptoms among IPV survivors who were garment workers living in urban slums from a previous study (Parvin et al., 2018), a sample size of 369 was required with a 95% level of significance and a .05 margin of error. We assumed a 10% non-response rate, given the high mobility of the slum population and the fact that data were collected from prior sampling frames of urban HDSS sites; thus, the final sample size was 405.

To separate the population with IPV experiences from the sample population, we preliminarily analysed and identified that 71.6% of women had experienced any form of IPV (physical, emotional and sexual violence) among the 405 women; therefore, we only included 290 women as the analytical sample for this study. However, it is worth noting that the observed 71.6% prevalence of IPV in the recruited HDSS sample should not be interpreted as a general prevalence estimate for all slum populations. Identifying women who experienced IPV is detailed in the *data collection tools and procedures* section.

### Data collection tools and procedures

A semi-structured questionnaire was used to collect response from each participant. The questionnaire was pretested on 50 women from non-selected slums in Dhaka city and modified based on participants' feedback. The questionnaire was developed with open and closed-ended questions related to sociodemographic characteristics, COVID-19 pandemic-related challenges, having non-communicable diseases (NCDs) and perceptions and attitudes about gender roles and household stress (Miller et al., 2010; Chien et al., 2018; De and Murshid, 2018; Chen and Harris, 2019; Gautam and H-S, 2019; Vilaplana-Pérez et al., 2020; Islam et al., 2021; Saud et al., 2021; Folayan et al., 2022; Koly et al., 2022; Sujan et al., 2023; Padmanabhanunni and Pretorius, 2024). The questionnaire also included two sections of the standardised WHO questionnaire on IPV and Post-Traumatic Stress Disorder checklist-5 (PCL-5). The WHO standardised questionnaire on IPV was used to identify the women with a history of IPV in the sample. The Bangla adaptation of the questionnaire included questions on physical, sexual and emotional violence inflicted by an intimate partner or husband. The participants were asked questions about their personal experiences with each episode of violence, as well as frequency. A five-point Likert scale (once, a few times, monthly, weekly or daily) was used to group individuals who answered "yes" to all IPV-related questions, thereby reducing the likelihood of recall bias (Palinkas et al., 2015) during data analysis.

The PCL-5 scale was utilised to screen PTSD symptoms in the sample population. The PCL-5, guided by the Diagnostic and Statistical Manual of Mental Disorders 5 (DSM-5), is a self-report measure consisting of 20 questions, previously validated and used in the Bangladeshi population (Islam et al., 2022). This helped determine the extent to which an individual was affected by trauma in the most recent month. The scores range from 0 (meaning "*not at all*") to 4 (meaning "*very much*"). The total number of elements is added to determine the severity of the subscale. The participants who scored higher than 31 were considered to have PTSD symptoms (Islam et al., 2022). In this study, the internal consistency reliability of the PCL-5 was 0.95, indicating excellent reliability.

### Data collection and participant identification procedure

For the data collection, female research assistants (RAs) received intensive training in conducting one-to-one interviews and collecting sensitive information about IPV and mental health. A detailed list of participants, including names, addresses and cell phone numbers, was provided to the RAs, which helped them locate the participants within the slum households. Following the consent procedures of previous IPV-related research and the ethical recommendations for studying violence against women (VAW) by the WHO (Naved et al., 2018), the RAs obtained oral consent from participants before data collection. The study objective and nature of the questions were shared with the participants, and their consent process was audio-recorded. RAs read the Bengali consent form and informed the participants about their right to withdraw from the study at any point without consequence, which can help them feel more at ease, reducing fear, subconscious resistance and concealment and ensuring participants feel more comfortable with their research involvement. The one-to-one interviews were conducted face-to-face in a private space preferred by the participant, at a time convenient to them. After completing the data collection phase, open responses were given post-codes.

### Data analysis

The results for participants' characteristics were evaluated by first categorising them according to their exposure to PTSD (yes/no) and then using descriptive statistics for continuous variables and percentage distribution for the categorical variables.

Explanatory variables were selected by a comprehensive literature search and matched with the available variables and proxies (Miller et al., 2010; Chien et al., 2018; De and Murshid, 2018; Chen and Harris, 2019; Gautam and H-S, 2019; Vilaplana-Pérez et al., 2020; Islam et al., 2021; Saud et al., 2021; Folayan et al., 2022; Koly et al., 2022; Sujan et al., 2023; Padmanabhanunni and Pretorius, 2024). Then, we tested those variables at a 10% significance level and incorporated them into the final model. The explanatory variables were

- **Sociodemographic characteristics:** age (continuous, in years), attended school (0 = no, 1 = yes; reference = attended school), working status (0 = non-working, 1 = working; reference = non-working), early marriage (0 = no, 1 = yes; reference = ≥18 years), having any NCD(s) (1 = yes, 0 = no; reference = no), husband's age (continuous, in years), husband's earning type (0 = non-earner, 1 = daily and 2 = monthly; reference = monthly), husband's substance use (1 = yes, 0 = no; reference = no), number of family members (1 ≤ 5 and 2 ≥ 5) and Crowding Index (Chien et al., 2018; Vilaplana-Pérez et al., 2020; Islam et al., 2021; Sujan et al., 2023). The Crowding Index was calculated based on the total number of individuals in each room of the residence.
- **COVID-19-related stressors:** We also used some COVID-19 stress-related variables that were found to be associated with IPV and PTSD during the literature search, namely: reduced income (1 = yes, 0 = no), worried about infection (1 = yes, 0 = no), isolation (1 = yes, 0 = no) and food availability (1 = yes, 0 = no) (Islam et al., 2021; Folayan et al., 2022; Sujan et al., 2023; Padmanabhanunni and Pretorius, 2024).
- **Behavioural factors:** Regarding behavioural issues, we considered: having arguments with husband (1 = more, 0 = less/same), justifying the participant's attitudes of wife beating (0 = negative, 1 = positive), participant's opinions being considered while making important family decision (1 = yes, 0 = no) and having a husband who often quarrels with others

(1 = yes, 0 = no) (De and Murshid, 2018; Saud et al., 2021; Koly et al., 2022).

- **Women's independence variables**: Regarding women's independence, we included dichotomous variables of whether: (i) husband restricts communication with family members, (ii) husband restricts communication with neighbours, (iii) husband restricts communication with others, (iv) husband restricts ability to go anywhere (market, neighbour's house) and (v) if they require husband's permission for healthcare visits (Miller et al., 2010; Chen and Harris, 2019; Gautam and H-S, 2019).

The level of PTSD was reported with a 95% confidence interval (CI) for all forms of IPV. We performed a crude logistic regression analysis as the primary method for selecting explanatory variables for the final model. Variables that became significant at a 15% significance level were considered in the multivariate logistic regression model. This relaxed threshold was chosen according to the purposeful selection approach and reduces the risk of prematurely losing potentially important associated factors, including confounders (Bursac et al., 2008). However, several variables, that is, women's age, education and husband's substance use, were emphasised in the multivariate model as these variables were found to be associated factors for mental health conditions in previous literature (Chien et al., 2018; Islam et al., 2021; Sujan et al., 2023). Variables that significantly contributed to PTSD in the multivariate model at different levels of significance (<0.1, <0.05, <0.01 and <0.001) were presented, and their adjusted odds ratios with 95% CIs were reported. The multivariate model included participant's age, education, husband's substance use and all variables with $p < .15$ from the bivariate analysis, including participant's working status, husband's age, husband's earning type, participants having NCDs, participant worried about COVID-19 infection, having argument with husband, attitude towards wife beating, participant's opinion in family decision-making and requirement of husband's permission to visit healthcare. All the analyses were conducted in STATA Windows version (15.1).

### Patient and public involvement

The study was conducted in accordance with the WHO ethical recommendations for the study of VAW (Naved et al., 2018), upholding participants' anonymity and privacy. The participants were informed about the study objectives, the voluntary nature of their participation and their full independence to discontinue participation in the research at any time. Moreover, the study team assured them of their privacy and the confidentiality of their information, noting that materials would be accessible only to the team and would be kept in a secure location. The data collection team scored the psychometric scale and shared information on free and low-cost psychosocial support services to those who scored high in PTSD. The research team involved in data collection and analysis received training on ethical conduct for research involving humans, as well as applicable personal and health privacy legislation.

### Author reflexivity statement

This study identified the prevalence and associated factors of PTSD among women who had experienced IPV during the COVID-19 pandemic, living in the slums of Dhaka city, Bangladesh. Since the study areas were part of an urban HDSS, the RA conducted door-to-door recruitment. KNK is a global mental health researcher based in Bangladesh and has prior experience conducting studies among women from slum communities. All study data were fully anonymised and are available at reasonable request from the Institutional Review Board secretariat of icddr,b.

## Results

### Participant's characteristics

A total of 290 (71.06%) women who experienced IPV during the COVID-19 pandemic were included for analysis. The average age was 30 years (95% CI: 29.5, 31.2), and nearly two-thirds (66.5%) attended school (see Table 1). Approximately 31% of women were daily earners working in clerical, administrative and managerial positions, as well as the ready-made garment sector. The majority of participants (87.9%) were married at an early age (before 18 years), with the average age of their husbands being 37.1 years (95% CI: 35.9, 38.2). More than half of the husbands (58%) were daily earners, followed by those who were monthly wage earners (36.2%). Around 66% of women reported that their husbands used substances such as jorda (boiled tobacco traditionally eaten with betel leaf), cannabis and other narcotics or drugs.

The average household size was 4.4, and more than 61% of women lived in families with <5 household members. About 58% of them reported having NCD(s), such as cardiovascular diseases (CVDs), diabetes and kidney diseases.

### Prevalence of PTSD among IPV survivors

Among the 290 participants who reported experiencing any form of IPV, about 21.16% ($n = 40$) had developed PTSD symptoms. In total, 25.74% ($n = 26$) of women who experienced all forms of IPV had developed PTSD (Figure 1). However, among the overall sample, 9.57% ($n = 11$) of women had PTSD while having no experience of IPV.

### Associated factors of PTSD

We conducted a bivariate logistic regression model with the possible explanatory variable, as presented in Table 2. Results show that participants who were working (1.81, 95% CI: 1.02, 3.20, $p < .05$), reported NCDs (3.05, 95% CI: 1.62, 5.75, $p < .01$); had older husbands (1.03, 95% CI: 1.00, 1.06, $p < .05$); husbands who were daily earners (1.91, 95% CI: 1.01, 3.61, $p < .05$) or were unemployed (3.89, 95% CI: 1.29, 11.73, $p < .05$) had increased arguments with their husbands (3.09, 95% CI: 1.75, 5.47, $p < .001$) and were found to be significant compared to counterparts (Table 3). Moreover, participants who reported that their opinion was not considered in family decisions (2.05, 95% CI: 1.11, 3.76, $p < .05$) were also more likely to have PTSD symptoms.

A multivariate logistic regression analysed the associated factors of PTSD among women who were exposed to IPV (Table 3). Women who reported having NCDs had 3.29 ($p < .05$) times more likelihood of having PTSD symptoms compared to those who did not have NCDs. In comparison to women who were not worried about COVID-19 infection, women who were worried were 3.87 ($p < .05$) times more likely to suffer from PTSD symptoms. Participants who had increased arguments with their husbands during the pandemic were three times ($p < .001$) more likely to have PTSD symptoms than those who did not.

**Table 1.** Participants characteristics

| Participant's characteristics | PTSD | Non-PTSD | Overall |
|---|---|---|---|
| Age* | 31.44 (29.57, 33.31) | 30.02 (29.01, 31.03) | 30.34 (29.46, 31.23) |
| Participants attending school | | | |
| No | 26 (26.8) | 71 (73.2) | 97 (33.5) |
| Yes | 40 (20.7) | 153 (79.3) | 193 (66.5) |
| Participant's working status | | | |
| Non-working | 39 (19.4) | 162 (80.6) | 201 (69.3) |
| Working | 27 (30.3) | 62 (69.7) | 89 (30.7) |
| Early marriage | | | |
| Yes | 59 (23.1) | 196 (76.9) | 255 (87.9) |
| No | 7 (20.0) | 28 (80.0) | 35 (12.1) |
| Husband's age* | 39.23 (36.50, 41.95) | 36.44 (35.22, 37.67) | 37.08 (35.94, 38.21) |
| Husband's earning type | | | |
| Monthly | 16 (15.2) | 89 (84.8) | 105 (36.2) |
| Daily | 43 (25.6) | 125 (74.4) | 168 (57.9) |
| Non-earner | 7 (41.2) | 10 (58.8) | 17 (5.9) |
| Husband's substance use | | | |
| Yes | 48 (25.1) | 143 (74.9) | 191 (65.9) |
| No | 18 (18.2) | 81 (81.8) | 99 (34.1) |
| Number of family members* | 4.21 (3.84, 5.58) | 4.46 (4.23, 4.69) | 4.40 (4.21, 4.60) |
| <5 | 44 (24.7) | 134 (75.3) | 178 (61.4) |
| ≥5 | 22 (19.6) | 90 (80.4) | 112 (38.6) |
| Participants having NCDs | | | |
| No | 15 (12.4) | 106 (87.6) | 121 (41.7) |
| Yes | 51 (30.2) | 118 (69.8) | 169 (58.3) |

*Note*: NCDs, non-communicable diseases; PTSD, post-traumatic stress disorder.
*Mean with 95% CI was reported.

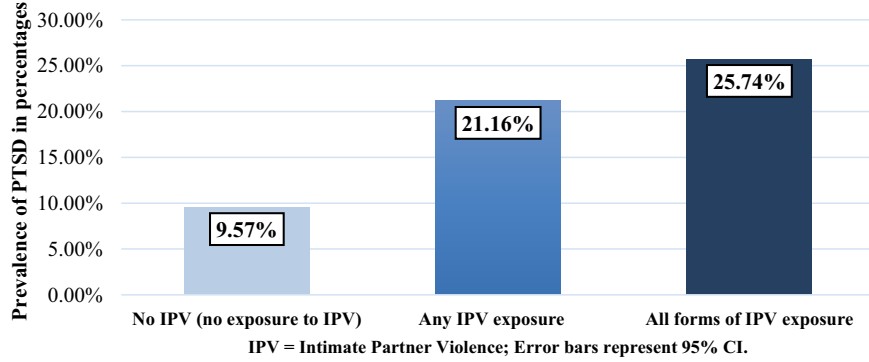

**Figure 1.** Prevalence of PTSD by IPV exposure status among women in urban slums. PTSD prevalence was 9.57% among women without IPV exposure, 21.16% among women exposed to any form of IPV and 25.74% among those exposed to all forms of IPV. Error bars represent 95% confidence intervals.

**Table 2.** Associated factors of PTSD among the women exposed to intimate partner violence (IPV) using the bivariate logistic regression

| Variables | COR | SE | 95% CI | *p*-value |
|---|---|---|---|---|
| Participant's age | 1.024 | .018 | .988–1.061 | .188 |
| Participants attending school | | | | |
| Yes | 1.0 | | | |
| No | 1.401 | .406 | .793–2.473 | .245 |
| Participant's working status | | | | |
| Housewife/non-working | 1 | | | |
| Working | 1.809 | .527 | 1.022–3.203 | .042* |
| Early marriage | | | | |
| No | 1.0 | | | |
| Yes | 1.204 | .539 | .500–2.897 | .678 |
| Husband's age | 1.028 | .014 | 1.001–1.056 | .046* |
| Husband's earning type | | | | |
| Monthly earner | 1.0 | | | |
| Daily earner | 1.913 | .62 | 1.014–3.611 | .045* |
| Non-earner | 3.894 | 2.191 | 1.292–11.73 | .016* |
| Husband's habit of substance use | | | | |
| No | 1.0 | | | |
| Yes | 1.51 | .467 | .824–2.77 | .183 |
| Number of family members | | | | |
| <5 | 1.0 | | | |
| ≥5 | .744 | .219 | .418–1.326 | .316 |
| Household crowding index | 1.131 | .146 | .878–1.455 | .341 |
| Participants having NCDs | | | | |
| No | 1.0 | | | |
| Yes | 3.054 | .986 | 1.622–5.75 | .001* |
| *Impacts of the COVID–19 pandemic* | | | | |
| Reduced income | | | | |
| No | 1.0 | | | |
| Yes | .813 | .44 | .282–2.348 | .702 |
| Participant worried about COVID–19 infection | | | | |
| No | 1.0 | | | |
| Yes | 2.397 | 1.323 | .813–7.07 | .113 |
| Having argument with husband | | | | |
| Same/less | 1.00 | | | |
| More than before | 3.09 | .9 | 1.746–5.467 | 0.000* |
| Isolated due to lockdown | | | | |
| No | 1.00 | | | |
| Yes | 1.377 | .549 | .630–3.009 | .423 |
| Food availability | | | | |
| Yes | 1.00 | | | |
| No | 2.035 | 1.133 | .684–6.058 | .202 |
| Participant's attitude of wife beating justified | | | | |
| Negative | 1.00 | | | |
| Positive | 1.348 | .412 | .740–2.454 | .329 |

(*Continued*)

**Table 2.** (*Continued*)

| Variables | COR | SE | 95% CI | p-value |
|---|---|---|---|---|
| Participants' opinions or decisions are being considered for the family decision | | | | |
| Yes | 1.00 | | | |
| No | 2.045 | .635 | 1.113–3.76 | .021* |
| Require husband's permission to visit healthcare | | | | |
| No | 1.00 | | | |
| Yes | 0.579 | .197 | .297–1.129 | .109 |
| Husband restricts communication with family members | | | | |
| No | 1 | | | |
| Yes | 1.542 | .735 | .606–3.925 | .363 |
| Husband restricts communication with neighbours | | | | |
| No | 1.00 | | | |
| Yes | 1.347 | .449 | .701–2.59 | .372 |
| Husband restricts and refrains from going anywhere (market, neighbour's house) | | | | |
| No | 1.00 | | | |
| Yes | 1.048 | .311 | .586–1.874 | .875 |
| Husband restricts communication with others | | | | |
| No | 1.00 | | | |
| Yes | 0.906 | .26 | .516–1.591 | .730 |
| Husband often quarrels with others | | | | |
| No | 1.00 | | | |
| Yes | 1.947 | 1.086 | .653–5.811 | .232 |

*Note*: CI, confidence interval; COR, crude odd ratio; SE, standard error.
Variables were considered statistically significant at *p* < 0.05 (95% CI). Statistically significant values are indicated with an asterisk (*). Reference categories are explicitly stated in each variable group.

**Table 3.** Associated factors of PTSD among the women exposed to IPV using the multivariate logistic regression (*n* = 290)

| Variables | AOR | SE | 95% CI | p-value |
|---|---|---|---|---|
| Participant's age | 0.97 | .039 | 0.895–1.046 | .405 |
| Participants attending school | | | | |
| Yes | 1.00 | | | |
| No | 1.21 | .465 | 0.569–2.569 | .622 |
| Participant's working status | | | | |
| Housewife/non-working | 1.00 | | | |
| Working | 1.84 | .646 | 0.928–3.666 | .081 |
| Husband's age | 1.02 | .031 | 0.961–1.083 | .510 |
| Husband's earning type | | | | |
| Monthly earner | 1.00 | | | |
| Daily earner | 1.48 | .545 | 0.718–3.043 | .289 |
| Non-earner | 2.68 | 1.761 | 0.741–9.711 | .133 |
| Husband's habit of substance use | | | | |
| No | 1.00 | | | |
| Yes | 1.55 | .537 | 0.786–3.058 | .206 |
| Participants having NCDs | | | | |
| No | 1.00 | | | |
| Yes | 3.29 | 1.196 | 1.613–6.709 | .001* |

**Table 3.** (*Continued*)

| Variables | AOR | SE | 95% CI | *p*-value |
|---|---|---|---|---|
| Participant worried about COVID–19 infection | | | | |
| No | 1.00 | | | |
| Yes | 3.87 | 2.426 | 1.134–13.218 | .031* |
| Having argument with husband | | | | |
| Same/Less | 1.00 | | | |
| More than before | 3.00 | .992 | 1.572–5.737 | .001* |
| Attitude of wife beating is justified | | | | |
| Negative | 1.00 | | | |
| Positive | 1.88 | .673 | 0.930–3.789 | .079 |
| Participants' opinions or decisions are being considered for the family decision | | | | |
| Yes | 1.00 | | | |
| No | 1.74 | .656 | 0.828–3.643 | .144 |
| Require permission for visiting healthcare | | | | |
| No | 1.00 | | | |
| Yes | 0.49 | .207 | 0.213–1.119 | .090 |
| No | 1.00 | | | |
| Yes | 1.56 | .949 | 0.471–5.141 | .468 |
| Husband restricts to communicate with neighbours | | | | |
| No | 1.00 | | | |
| Yes | 1.38 | .586 | 0.602–3.171 | .446 |
| Husband restricts to go anywhere (market, neighbour's house) | | | | |
| No | 1.00 | | | |
| Yes | 1.31 | .494 | 0.629–2.747 | .466 |
| Constant | **0.02** | **.017** | **0.002–0.128** | **.000*** |

*Note*: Model fit-$\chi^2$: 52.708, $p$ < .001, pseudo-$r^2$: 0.169.
AOR, adjusted odds ratio; CI, confidence interval; SE, standard error.
Variables were considered statistically significant at $p$ < .05 (95% CI). Statistically significant values are indicated with an asterisk (*). Reference categories are explicitly stated.

## Discussion

PTSD is one of the leading causes of psychosocial disability, impaired social functioning and low health-related quality of life (Miethe et al., 2023). Additionally, IPV is identified as one of the emerging risk factors of PTSD worldwide, being a significant public health and human rights issue (Miethe et al., 2023). Research has shown that women are more susceptible to both PTSD and IPV, pointing towards a bidirectional relationship between the two factors (Kemp et al., 1991; Miller et al., 2010; Chien et al., 2018; De and Murshid, 2018; Fernández-Fillol et al., 2021; Sekoni et al., 2021; Langhinrichsen-Rohling et al., 2022; Lyons and Brewer, 2022). Furthermore, increased incidence of IPV during the COVID-19 pandemic might lead to significant psychological impacts, such as PTSD in women globally, including low and middle income countries (LMICs), and also in Bangladesh (Whittle et al., 2019; Islam et al., 2021; Rashid Soron et al., 2021; Folayan et al., 2022; Uzoho et al., 2023).

Historically, the rate of IPV is higher in low-income communities, such as rural–urban migrants living in slums, and its mental health impact is well-documented in international studies (Dutton, 1995; Brewin et al., 2000; Maercker and Müller, 2004; Fernández-Fillol et al., 2021). In terms of Bangladesh, the laws for preventing VAW give special attention to dowry-related violence, rape,

murders, and so forth. However, it does not recognise all forms of violence, including IPV, and ignores the prime influential factors such as the influence of in-laws, husbands, socio-economic contexts and religious sanctions. Despite such vulnerabilities, people living in Bangladesh have low mental healthcare-seeking tendencies and a high level of stigma, with no community-based mental health services, especially for underprivileged communities (Nuri et al., 2018; Hasan et al., 2021). Although 96.3% of the domestic violence survivors in Bangladesh believe mental health support is necessary, only 25% are aware of the available services and know how or where to access them (Rashid Soron et al., 2021). As a result, IPV and its mental health consequences remain unaddressed. Therefore, our study is the first of its kind to explore the prevalence of PTSD and its determinants in a vulnerable population, such as women living in urban slums who were exposed to IPV (Dutton, 1995; Brewin et al., 2000; Maercker and Müller, 2004; Fernández-Fillol et al., 2021). In Bangladesh and the Global South, women living in slums are excellent drivers of the informal sector, contributing to their households, communities and the country's economy. Thus, safeguarding their well-being is not only critical for their human rights and personal health but also essential for the overall socio-economic stability and development of these regions (Adams et al., 2015; Al Helal et al., 2017).

Our study reported that about 21.16% of women who experienced IPV had developed PTSD symptoms in the selected urban slums of Dhaka. The relatively high rate of PTSD might be due to poor living situations, low socio-economic status, lack of accessibility to basic resources, vulnerability to infectious diseases and lack of social support (Fatemeh et al., 2021). Additionally, a systematic review has indicated that the majority of the (31–84.4%) women with exposure to IPV suffered from PTSD (Woods et al., 2008). Research has also shown that women with a history of IPV reported a significantly higher rate of PTSD symptoms as compared to women who were not exposed to IPV (Pico-Alfonso, 2005). Living with an abusive partner can exacerbate symptoms associated with PTSD, as they can experience a chronic feeling of fear and stress (Pico-Alfonso, 2005). Such cases are highly prevalent in LMICs like Bangladesh, where accessibility to mental health services is low, and where women living in slums face low education, early marriage, poor wealth indices and patriarchal gender norms (Gunarathne et al., 2023).

Worldwide, different studies supported the association between the presence of NCDs, such as CVDs, diabetes mellitus (DM), hypertension, kidney disease and PTSD (Mulugeta et al., 2019; Stein et al., 2019). This is consistent with our findings, where the women who self-reported having any form of NCDs were found to be 3.29 times as likely to have PTSD symptoms, which might be due to their limited ability to manage complex health issues. Furthermore, for people living in slums, accessing healthcare is more challenging in urban areas than in rural areas due to the lack of primary healthcare infrastructure (Adams et al., 2015). The slum community-based urban satellite clinics found in some urban areas are typically temporary, project-based and do not prioritise mental health care, leaving public tertiary care facilities as the only option for people living in urban slums. These facilities are seldom accessed due to long distances and high traffic, which deters people from low-income communities from seeking timely care. They often neglect their conditions out of fear of catastrophic out-of-pocket expenses associated with private care (Afsana and Wahid, 2013; Mannan, 2013; Adams et al., 2015; Al Helal et al., 2017; Ahamad, 2020). In addition to already being susceptible to common mental health problems such as stress and anxiety due to living in precarious conditions, the management of long-term chronic NCD conditions may intensify distress among women. Exposure to IPV may exacerbate psychological distress, resulting in elevated PTSD symptoms (Stein et al., 2019; Jones and Gwenin, 2021), as elucidated by the WHO five-by-five approach that explains the bidirectional relationship between NCD and PTSD (Stein et al., 2019). Prolonged periods of stress can lead to elevated cortisol levels in the bloodstream and hypothalamic–pituitary–adrenal axis dysregulation, leading to a wide range of NCDs like DM, CVD and autoimmune diseases (Jones and Gwenin, 2021). Therefore, in developing settings like Bangladesh with a dearth of mental health specialists, collaborative care models have the potential to combat the double burden of chronic NCDs and mental health conditions.

Interestingly, during the COVID-19 pandemic, mental health conditions, including PTSD, were exacerbated globally as a result of stressors like fear of infection, food unavailability, income loss and social distancing measures, particularly for people in the lower-income group (Buttell and Ferreira, 2020; Agüero, 2021; Zhu et al., 2021; Langhinrichsen-Rohling et al., 2022). These challenges also increased the incidence of IPV during the pandemic (Buttell and Ferreira, 2020; John et al., 2020; Agüero, 2021; Zhu et al., 2021; Lyons and Brewer, 2022; Affairs UDoV PTSD, 2024). Evidence supports an association between pandemic-related worry and PTSD due to recurring negative thoughts and helplessness in a compromised situation (Jones and Gwenin, 2021; Zhu et al., 2021; Lyons and Brewer, 2022). Furthermore, worrying about COVID-19 may lead to insomnia (Koly et al., 2021; Lyons and Brewer, 2022). According to DSM-5, insomnia is one of the predisposing, precipitating and perpetuating factors of PTSD (Besedovsky et al., 2019; Garbarino et al., 2021), which could be linked to lower productivity, making the underprivileged women more susceptible to health conditions and related traumatic experiences (Besedovsky et al., 2019; Garbarino et al., 2021). This is consistent with our findings that women worrying about COVID-19 infection were about 3.87 times more likely to be displaying symptoms of PTSD.

Our study also reported that women who had increased arguments with their husbands during the pandemic were three times more likely to display PTSD symptoms than those who had their usual arguments with their partners. Several studies showed an increase in conflict between partners during a health emergency, which could lead to IPV and was further linked to economic insecurities and difficulties in accessing basic needs and social support (Langhinrichsen-Rohling et al., 2022; Ozad et al., 2022). Increased arguments between spouses may add to stress, sleep difficulties and loneliness, leading to the development of PTSD symptoms (Langhinrichsen-Rohling et al., 2022; Ozad et al., 2022).

This study highlights the syndemicity of PTSD and IPV among Bangladeshi urban women from low-income settings. The vulnerability of these women to IPV interplayed with the poor health and environmental conditions of slums and pandemic-related issues, triggering the symptoms of PTSD and exposure to IPV. To prevent IPV-related PTSD, policymakers should prioritise screening services for IPV and mental health conditions in primary health care facilities, along with relevant psychosocial support for the survivors, which was found promising elsewhere (Iskandar et al., 2014). These findings have the potential to inform the development of policies to train the lay health workforce in addressing IPV-related mental health impacts, and to make necessary amendments and implement related laws. Additionally, they can help increase community-based awareness activities in low-income areas, such as slums.

Furthermore, it can inform our understanding of factors associated with PTSD in women exposed to IPV, as well as the impact of the slum environment on the mental health of residents, helping formulate appropriate strategies to prevent IPV and other traumatic experiences in such marginalised groups. Based on our findings, it is also crucial to understand the factors that lead to spousal arguments and implement community-based interventions that involve both males and females to raise awareness of IPV and its negative impacts. Local authorities and policymakers can mitigate interspousal conflicts during health crises by promoting resilience and coping mechanisms through awareness campaigns, particularly those tailored to slum communities. Despite strong associations found in past studies of factors like occupational status and substance use, these were not significant contributors to PTSD in our sample population (Carbone et al., 2019; Radcliffe et al., 2021), which could be due to multiple factors, such as participants possibly sharing more socially acceptable answers rather than accurate descriptions, leading to reporting bias. Since this study examined a socially stigmatised topic among a vulnerable population, it may have contributed to social desirability bias. Considering that this was a cross-sectional study, we cannot make causal inferences. The small sample size may not have been sufficient to detect a difference between the two groups, limiting generalisability to all Dhaka slums.

Additionally, as both IPV and PTSD are highly sensitive topics in the Bangladeshi socio-cultural context, there is a possibility of reporting bias. Participants may have under-reported their experiences of violence or psychological symptoms due to fear, stigma or social desirability, which could have led to underestimation of the true prevalence. Future studies should employ a larger sample size and select participants from women who have already been medically screened for IPV to reduce social desirability and subjective bias.

Despite these limitations, our study has some significant strengths. To the best of our knowledge, this study generated baseline evidence of PTSD and its determinants in women living in slums who experienced IPV in Bangladesh and expanded the scarce body of literature related to IPV and mental health conditions. In particular, the study engaged a hard-to-reach population, ensuring that their risks and challenges were represented in research. Drawing attention to the high prevalence of PTSD among IPV victim-survivors, and the potential association can support policymakers to endorse related interventions for this marginalised population. Findings further indicated the need to conduct large-scale studies to identify risk factors of PTSD in women with a history of IPV in Bangladesh. Since the population is part of a surveillance system, interventional studies can be conducted to explore the behavioural changes of this population. Finally, this study can serve as a reference for devising strategies and policies to improve the living conditions and mental health of slum dwellers.

## Conclusion

The study is the first of its kind to be based on a marginalised population of Bangladesh that aimed to identify associated factors of PTSD in IPV-exposed women. The study recommends systematic early PTSD screening, which might facilitate the detection of those who need urgent and continuous support. Findings could guide policymakers in achieving Sustainable Development Goal 3 by reforming the existing urban healthcare system to include mental health support, thereby promoting the well-being of vulnerable groups in Bangladesh.

**Open peer review.** To view the open peer review materials for this article, please visit http://doi.org/10.1017/gmh.2025.10092.

**Supplementary material.** The supplementary material for this article can be found at http://doi.org/10.1017/gmh.2025.10092.

**Data availability statement.** The full data for this study are not available in a public repository due to ethical restrictions. Also, it is not permissible under the Institutional Review Board of icddr,b to share data in public repositories, as this could result in a serious breach of the ethical rights of the study participants. During the consenting process, the participants were informed that their data would be managed only by the research team, stored in a secure place accessible only to the research team and destroyed after publication. We ensured participants about the confidentiality and protection of their personal information. Therefore, any requests regarding data availability can be sent to M. A. Salam Khan, Institutional Review Board (IRB) Coordinator of icddr,b (Address: IRB Secretariat, Research Administration, icddr,b, Mohakhali, Dhaka-1212).

**Acknowledgements.** The authors would like to express their deepest gratitude to all respondents who participated in this study. The authors would like to give special thanks to Dr. Sabrina Rasheed, Scientist at the International Centre for Diarrhoeal Disease Research, Bangladesh (icddr,b), for her valuable insights while conceptualising this project. icddr,b is grateful to the Governments of Bangladesh and Canada for providing core/unrestricted support.

**Author contribution.** K.N.K. and J.S. conceptualised the study. K.N.K. and J.S. designed the study. K.N.K. and M.A.B. led the recruitment and formal analysis. S.S., M.K.M., J.S. and M.A.B. prepared the manuscript, and K.N.K. supervised the study, analysis and writing process. The draft manuscript was reviewed and edited by K.N.K., J.W. and B.B.N. All authors contributed to the intellectual content of the manuscript and read and approved the final version of the manuscript.

**Financial support.** This project is funded and supported by the Mujib 100 Research Grants for Women (Mujib 100 RGfW) by icddr,b.

**Competing interests.** The authors declare none.

**Patient and public involvement.** Patients and/or the public were involved in the design or conduct or reporting, or dissemination plans of this research. Refer to the Methods section for further details.

**Patient consent for publication.** Consent obtained directly from patient(s).

**Ethics approval.** Both the Research Review Committee (RRC) and Ethics Review Committee (ERC) of the Institutional Review Board (IRB) of the icddr,b approved this study [PR-22001].

**Provenance and peer review.** Not commissioned; externally peer reviewed.

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
