## [Reviewer Report]

Title: In title PTSD can be abbreviated.

Abstract: The Abstract is clear.

Introduction: The Introduction builds a solid background, but some modifications will make the novelty and rationale more prominent and directly link the study to an identifiable evidence gap. To strengthen the research gap related section, I recommend:

• Consider shortening background details to focus on the study gap and rationale.

• Moving this statement earlier in the final paragraph and expanding it to specify that prior Bangladeshi studies on IPV have focused mainly on prevalence, depression, or anxiety, with PTSD largely overlooked.

• Highlighting that there is no evidence on how COVID-19-related stressors interact with IPV to influence PTSD in urban slums, despite these being high-risk environments.

• Explicitly noting the absence of studies using validated PTSD assessment tools (e.g., PCL-5) in this population.

Methods: The methodology is described in detail. However, the explanation of the sample reduction from 405 to 290 IPV-exposed participants should be clarified. Please also clarify that the reported 71.6% IPV prevalence was observed in the recruited HDSS sample and is not a general prevalence estimate for all slum populations, to avoid misinterpretation of representativeness. In the Patient and Public Involvement section, please provide a formal citation to the WHO ethical and safety recommendations for research on violence against women consistent with the referencing style used elsewhere in the manuscript.

Issues with the explanatory variable section include:

• Mixed variable types without grouping – Sociodemographic variables (age, education, working status, early marriage, NCD status, husband’s age, etc.) are listed alongside COVID-19 stress variables, behavioural variables and women independence related, without clear grouping or subheadings, making it hard to see which are demographics, which are COVID-related, and which are IPV-related attitudes/behaviours.

• Inconsistent coding explanation – Some variables have coding explained (e.g., “0=No, 1=Yes”), others do not (e.g., “husband’s age”), and some reference categories are not explicitly stated.

• Crowding index definition placed mid-list – This disrupts flow; consider defining it separately or in the Measures section.

• Additionally, consider defining the Crowding Index separately for clarity. Additionally, please provide a brief justification or methodological reference for using p < 0.15 as the threshold for selecting variables from the bivariate to the multivariate analysis. This will enhance transparency and allow readers to better understand the rationale behind your modelling approach.

Results: The results are clearly presented, but –

• Tables 2 and 3 require consistent p-value formatting and explicit indication of reference categories. Figure 1 could be made more self-explanatory. Figure 1 labels could be made more self-explanatory for readers without referring to the text. Additionally, this figure should identify which part is for IPV-exposed and which part is non-exposed.

• Additionally, there is an inconsistency in how education is described. In the Data Analysis section, it is coded as ‘attended school (0=No, 1=Yes)’, while in Table 2 it is presented as ‘educated’ vs ‘non-educated’. Please ensure terminology is consistent and define the exact criteria used for categorisation.

• For Table 3, please explicitly state in the table legend or footnote which variables were included in the adjustment for the AORs. Currently, readers have to infer from the Methods that the model included age, education, husband’s substance use, and variables with p < 0.15 from the bivariate analysis; listing them directly in the table would improve clarity.

Discussion: The discussion appropriately interprets findings in the context of global literature, but some organisational pattern can be considered that is separating prevalence interpretation, determinants discussion, and policy implications into distinct paragraphs for better flow. Moreover, some sentences imply causality between IPV and PTSD despite the cross-sectional design The statement “Living with an abusive partner can exacerbate trauma, leading to PTSD” is supported by reference, but that study was also observational. As your own study is cross-sectional, it would be more accurate to frame this as “associated with PTSD” or “linked to higher PTSD symptoms,” and attribute the causal phrasing to prior research rather than implying causality from your findings.

Typographical corrections:

• Replace ‘at at’ with ‘at a’ in the Methods section.

• Correct ‘utlised’ to ‘utilised’ in the Study setting description.

• Change ‘husband that often quarrels’ to ‘husband who often quarrels’ for grammatical accuracy.

• Ensure consistent hyphenation for ‘non-exposed’.

• Standardise percentage formatting (e.g., ‘21.2% (n=40)’ rather than ‘21.2 % (n=40)’).

• In the discussion, “slumsFuture” should be corrected to “slums. Future”.

• Review reference formatting for consistency, ensuring all DOIs are preceded by ‘https://doi.org/’.

---

## [Reviewer Report]

Dear Authors,

Thank you for the opportunity to review this manuscript. The study makes a timely and important contribution by addressing a clear gap in global mental health research: the prevalence and determinants of PTSD among women experiencing intimate partner violence in Dhaka’s urban slums during the COVID-19 pandemic. The manuscript is well-organized, methodologically sound, and provides a comprehensive examination of a highly vulnerable population.

Major Strengths:

1. Relevance and originality – The focus on urban slum women in Bangladesh fills an important evidence gap. This is the first study of its kind in this context, and the findings have practical policy implications.

2. Methodological rigor – The use of a validated Bangla version of PCL-5 and WHO IPV questionnaire adds robustness. Clear inclusion/exclusion criteria and detailed description of ethical safeguards strengthen the credibility of the study.

3. Policy and practice implications – The discussion connects well with the broader literature and highlights how findings could inform community-based interventions and health system responses.

I believe the manuscript is suitable for publication after minor revisions. Below are my detailed comments and suggestions:

Suggestions for Minor Revision

1. Methods: The methods are solid overall. The authors use validated tools (WHO IPV questionnaire, Bangla PCL-5), and the ethical considerations are well thought out, which is critical for such a sensitive topic. The analysis is appropriate, but I’d like a bit more clarification on the sample size issue — they calculated for 405 participants but the analysis included 290 IPV survivors. This doesn’t undermine the study but should be discussed more explicitly as a limitation.

2. Presentation: The paper is generally well written and structured. That said, there are a few areas where clarity could be improved:

- PTSD prevalence is reported slightly differently in places (22.6% vs. 21.2% among survivors), so the numbers should be made consistent.

- Figure 1 is helpful but the labels are a bit confusing (“any form of IPV” vs. “all forms of IPV” could be explained more clearly).

- The discussion is thorough but at times leans heavily on global literature. I’d encourage the authors to balance this with more references from South Asia or LMICs to ground the findings in similar contexts.

- The limitations are covered well, but I’d suggest adding a note about possible reporting bias given the sensitive nature of IPV and PTSD.

- A light language edit would help tidy up some redundancies (e.g., repeated phrasing about “vulnerable populations” in discussion).

Recommendation: Overall, I think this is a strong paper that deserves publication. My recommendation is accept with minor revisions. The revisions are mostly about consistency in reporting, clarifying some details, and improving presentation, not about the core science, which is already solid.